# Broadband highly directive 3D nanophotonic lenses

Eric Johlin[1,2], Sander A. Mann[1,3], Sachin Kasture[1], A. Femius Koenderink[1] & Erik C. Garnett[1]

Controlling the directivity of emission and absorption at the nanoscale holds great promise for improving the performance of optoelectronic devices. Previously, directive structures have largely been centered in two categories—nanoscale antennas, and classical lenses. Herein, we utilize an evolutionary algorithm to design 3D dielectric nanophotonic lens structures leveraging both the interference-based control of antennas and the broadband operation of lenses. By sculpting the dielectric environment around an emitter, these nanolenses achieve directivities of 101 for point-sources, and 67 for finite-source nanowire emitters; 3× greater than that of a traditional spherical lens with nearly constant performance over a 200 nm wavelength range. The nanolenses are experimentally fabricated on GaAs nanowires, and characterized via photoluminescence Fourier microscopy, with an observed beaming half-angle of 3.5° and a measured directivity of 22. Simulations attribute the main limitation in the obtained directivity to imperfect alignment of the nanolens to the nanowire beneath.

[1] Center for Nanophotonics, AMOLF, Science Park 104, 1098 XG Amsterdam, The Netherlands. [2] Department of Mechanical and Materials Engineering, University of Western Ontario, 1151 Richmond St., London, ON N6A 3K7, Canada. [3] Present address: Photonics Initiative, Advanced Science Research Center, City University of New York, 85 St Nicholas Terrace, New York, NY 10031, USA. Correspondence and requests for materials should be addressed to E.J. (email: johlin@alum.mit.edu) or to E.C.G. (email: garnett@amolf.nl)

An emitting or absorbing system can be characterized by the response at one angle relative to the mean response over all angles, defined as the directivity, as depicted in Fig. 1a. The ability to control the directivity of light is highly attractive for a number of applications—for example, improving out-coupling and direction of emission from single-photon emitters could improve their brightness by a factor of 3[1]; matching a nanostructured photovoltaic device's absorption to the solid angle of the sun promises to yield improvements of up to 278 mV in open-circuit voltage[2,3]; and lenses for nanoscale light-emitting diodes and lasers could improve the routing of light into specific directions of interest even over broad spectral ranges. The redirection of light has been achieved using macroscopic lenses for millenia, and is usually well described by a ray- or geometric-optics approximation, understood by the simple refraction of light at interfaces of changing permittivity[4].

Due to the advent of nanoscale sources and detectors of light, nanoscale and microscale structures to control the distribution of their emission/absorption have become of great interest. Here, we can largely distinguish between two different approaches: first, resonant antennas with size features on the order of the wavelength; and second, microscale lens-like structures with feature sizes of multiple wavelengths, designed within a ray-optics framework.

Nanoscale antennas work largely or exclusively through interference effects, allowing the antennas to cancel out propagation into certain directions, thereby creating a dramatic angular redistribution of emission or collection[5–7]. Furthermore, the existence of structural elements in the near-field of the emitter/receiver allows these structures to potentially modify the local density of optical states experienced by this active element, thereby changing the rates and efficiencies of optical interactions (e.g., absorption, or radiative out-coupling), and enhancing the brightness of a source[8–11]. Additionally, these near-wavelength structural elements can collect light from regions that exceed the physical extent of the structure, as is observed in many nanostructured materials[5,12,13]. Although nanoscale antenna arrays have shown simulated directivities of ~25[6], and forward to backward (F/B) ratios of ~5.6[5], these responses often fall off rapidly within a narrow (~40 nm wavelength) bandwidth. These structures are usually made from metals[5–7,14,15] or high-index dielectric materials, most notably silicon[16–19], which generally limits the design to two-dimensional (2D) or simple three-dimensional (3D) structures that can be produced by controlled etching.

In contrast, micron-scale features are frequently employed to achieve efficient out-coupling for solid-state quantum light sources through a solid immersion lens[20–23], or can be used to create microlenses for directive emission[24–27]. These structures are readily designed following a simple ray-tracing approach, considering that the features of the structure are at least on the order of many wavelengths, and their performance is fully determined by the size of the structure[28], as the wave nature of light is not utilized. Recent developments in the field of integrated photonics, however, have demonstrated that algorithmic design of optical components, using a large number of spatial degrees of freedom, can lead to extremely efficient components with a very compact footing[29,30]. These resulting structures appear unintuitive; their functionality is hard to predict given their shape, owing to the fact that for small spatial features light is dominated by its wave nature.

Herein, we explore the design, simulation, and experimental validation of highly directive 3D nanophotonic lenses. We apply algorithmic design to create structures with a large number of spatial degrees of freedom, comprised of low-index dielectric ($n \approx$ 1.4) material. We find that such nanolenses occupy a beneficial

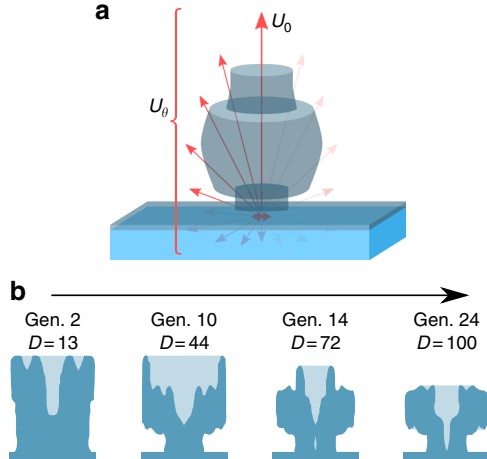

**Fig. 1** Diagram of directivity and optimization. **a** Schematic depicting the emission (arrows) from a point emitter and nanolens, with $U_0$ representing the emission in the $\theta = 0$ direction, and $U_\theta$ being the total emission in all angles. **b** Series of snapshots in an example optimization process, showing the revolved cross-sections (dark blue, with background material in light blue) of lens structures as the optimization progresses, accompanied by higher achieved directivity values, $D$

position on the characteristic spectrum of directive structures— while the structure is not resonant and the total size is multiple wavelengths, the critical features are sub-wavelength and enable effective utilization of nanophotonic interference effects to achieve large directivities over a wide bandwidth, while simultaneously improving out-coupling of light from the substrate. As we will show, this allows the algorithmically designed nanolens to significantly outperform traditional intuition-based lens designs of the same size and material. Experimental nanowire–nanolens systems are then fabricated, showing the emission transformed from nearly isotropic photoluminescence (PL) from bare wires, to beaming into a narrow solid angle after the nanolens is applied. Finally, optical simulations are employed to understand the difference between the measured and simulated directivity.

## Results

**Nanolens design.** The general philosophy of this work is to utilize algorithmic design to mold the dielectric environment around nanoscale emitters, reshaping their emission from isotropic to highly directive. We constrain the design space explored by the algorithm to structures and materials in immediate reach of state-of-the-art 3D nanolithography. In particular, fully 3D structures with features of order 100 nm can be directly written in low-index material by multiphoton polymerization[31]. Such low-index materials are actually a very beneficial choice for nanolenses due to their reduced parasitic absorption, as well as excellent processing compatibility with extant optoelectronic devices, as they are formed at room temperature and without aggressive chemical processing. Even within this search-space limitation, a huge number of spatial degrees of freedom can be optimized for nanophotonic lens design.

**Evolutionary algorithm.** An evolutionary algorithm (EA) is one popular approach to algorithmic design where a subset of best solutions from one generation of structures (parents) are combined to create the next generation. Such algorithms often allow convergence of an optimal solution significantly more efficiently than brute-force techniques, and are attractive when no functional form of the solution exists, a large number of degrees of freedom are being optimized, and when the design is being

performed using binary variables (e.g., the presence or absence of a fixed material, not a continuum of possible materials) making derivatives of solutions not readily available (or their calculation in a high-dimensional space unfeasible)[32].

EAs can be particularly attractive when designing physical structures for high performance in a certain desired output characteristic. They have been used previously to design 2D optical structures including metal and dielectric scatterers[33,34], plasmonic particle arrays[35], and linear antennas[36]. In contrast, herein we use an EA to create 3D, glass-like dielectric structures to sculpt the emission from nanoscale light sources giving them new functionality (enhanced directivity). The structures are smoothed through convolution with the asymmetric experimental lithography system point spread function, and translated into finite-difference time-domain (FDTD) simulations to compute the performance of the nanolens structures combined with emitter/absorber materials on substrates. After a number of generations goes by with little improvement, the design is considered converged. For more details on the specifics of the process, see Methods section.

**Nanolens structure.** Our nanolens is designed to tailor the dielectric environment of horizontal 80 nm diameter gallium arsenide (GaAs) nanowires, which are of particular interest due to their reasonably high brightness, and relevance to photovoltaics[37–40], nanoscale lasers[41], photodetectors[42], and other photonic technologies[43] where directional emission or absorption are highly attractive. Importantly, the PL from the nanowires comes mainly from a small ~240 nm junction region. Due to this relatively confined emission, and to aid in computationally efficient optimization, we constrain the lens to be axisymmetric, enabling us to optimize a 2D structure (the cross-section of the lens). However, by revolving the 2D structure around the center of the active element, we utilize 3D calculations of performance. This additionally permits the design of a lens that works both for point emitters (as is often of interest for single-photon emitters), as well as finite emitters with reasonably localized emission (e.g., axial junction nanowires)

An example of the progression of one design run is shown in Fig. 1b, with four snapshots of structure cross-sections shown at various points (generations) during the process. It should be noted that often there are two or more converged designs with similarly high directivities, and two or more designs with similar geometries between separate (non-interacting) optimizations. This is particularly useful as the former allows the design more suitable for fabrication to be selected, while the latter supports the supposition that the optimization was indeed well converged.

**Computational characterization.** One of the best performing structures for enhancing directivity is shown in Fig. 2. The performance of the lens is estimated through simulation of point emitter recombination at the center of the nanowire, first without the nanolens structure present (as depicted in Fig. 2a). The emission diagram of the nanowire alone (embedded in polymer index matched to the glass substrate) is plotted in Fig. 2b. The vector from the origin to a point on the plotted surface represents the relative emission into that direction. The upper blue region of the plot indicates emission into free space out of the substrate, whereas the lower red region indicates emission into the glass substrate. Directivity is calculated from such emission datasets, and defined simply as the flux emitted into a given angle, normalized by the mean emitted flux:

$$D(\phi, \theta) = \frac{4\pi U(\phi, \theta)}{\iint\limits_{S} U \sin(\theta)\, \mathrm{d}\phi\, \mathrm{d}\theta}, \qquad (1)$$

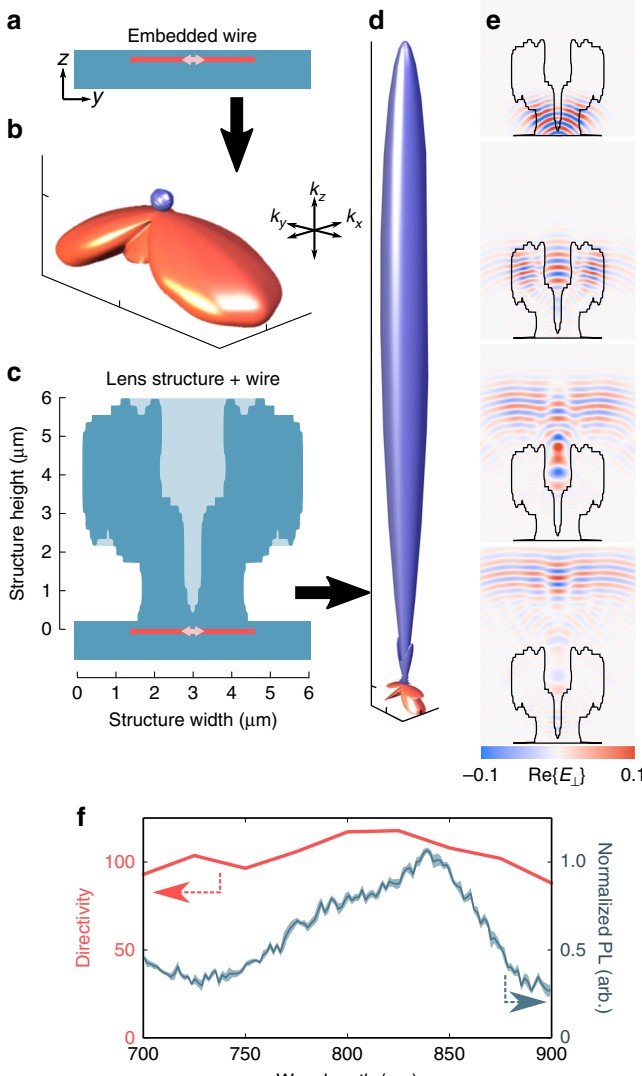

**Fig. 2** Nanophotonic lens performance. **a** Cross-sectional diagram of a 80 nm diameter, 3 μm long GaAs nanowire (red) with 250 nm emitting region (pink), embedded under 200 nm of polymer, on glass. **b** Normalized emission plot of dipole emission originating from the center of the nanowire. The upper (blue) region corresponds to emission into the upper hemisphere, whereas the lower (red) region corresponds to emission into the substrate, expressing a directivity of 0.75. **c** Cross-sectional schematic of the nanowire with the nanophotonic lens present. **d** Emission diagram from the nanowire–nanolens system, showing an enhanced directivity of 101. The two emission diagrams show the same total integrated emission within 5%. **e** Simulated propagation of normalized light emission through a nanolens structure (outlined, red) in four time snapshots, showing the presence of reflection and refraction (snapshot 1–3), and interference (2–4) effects. **f** Directivity into the surface normal as a function of wavelength, overlaid with the photoluminescence spectrum of the nanowire emitters used in the experiments, showing broadband response over the nanowire emission range (small shaded region is the standard deviation over 10 sample points)

were $U$ is the radiated flux into a given angle ($\phi, \theta$), depicted in Fig. 1a, and $S$ denotes the spherical surface of integration over all solid angles[44]. The directivity of the bare nanowire is calculated to be 0.75 in the surface normal direction, indicating that the nanowire sends a below-average amount of its emission directly out of the substrate.

The algorithmically designed structure is depicted in cross-section in Fig. 2c, and the resulting emission from the center of the nanowire–nanolens system is shown in Fig. 2d. The difference in profile before and after the lens is added is clear—the emission is transferred significantly into the surface normal ($z$) direction, while being attenuated in all other directions, with the total integrated emission remaining constant before and after nanolens application (within 5%). Quantitatively, this corresponds to the directivity (in the direction of the surface normal) for point-emission increasing from $D = 0.75$ to 101 when the nanolens is applied, with a F/B ratio of 46.5. Importantly, this directivity is highly broadband, remaining essentially constant over the PL range of GaAs, from 750 to 900 nm, as shown in Fig. 2f. It should be noted that the nanolens improves the emission out of the substrate as well. We calculate that the emission into the upper hemisphere increases by a factor of 2 (from 18.8 to 36.7%) when the nanolens is applied. As the total emission remains essentially constant, this indicates that the lens is not only improving the directivity of the light already being emitted out of the substrate, but shifting the emission from inside the substrate toward free space as well. This also allows an increased fraction of the emission from the nanowire to be collected even for low power objectives; for example, in a numerical aperture (NA) of 0.7, the collected emission increases by a factor of 2.8 (from 9.17 to 25.8%). The nanolens can thus be thought of as sculpting the emission from the nanowire.

The mechanism for the directive lensing is clarified in Fig. 2e (as well as in Supplementary Movie 1, available online), displaying a progression of the electric field (real part of the field component perpendicular to the plot) propagation from 850 nm peak emission (bandwidth of 194 nm) from the center of the nanowire over a 60 fs timespan (~20 fs between frames), experiencing reflection and refraction by the nanolens-air interfaces, as well as interference in the central cavity of the nanolens, and finally emission as a nearly planar wave upward into free space. It appears that it is this combination of reflection, refraction, and interference that allows the nanolens structures to maintain highly directive lensing over a broad wavelength range.

As a control, we compare the nanolens designed here with the performance of a classical (intuitively) designed spherical lens[16,22,28,45]. We find that for the same source, an optimally placed spherical lens of the same refractive index and radius as our nanolens (2.8 μm) leads to a directivity of 36.6 and F/B ratio of 7.2, with 21.6% of light being emitted out of the substrate. This indicates that a performance enhancement of 3× for directivity and 6× for the F/B ratio is expected by moving from intuitive to algorithmic design of nanolens structures, while extracting 70% (rel.) more light from the substrate.

**Experimental realization**. The fabrication of the nanowire–nanolens system begins by drop-casting 80 nm diameter GaAs nanowires onto clean glass coverslips. After drop-casting, the average nanowire length is approximately 6 μm. The nanowires are then coated with a protective SU-8 polymer layer, with a thickness of ~200 nm. This polymer spacer layer ensures that the nanowire is not exposed to the focused laser spot during the nanolens writing step, preventing laser damage.

Nanophotonic lens structures are then written directly onto the coated nanowire sample using two-photon absorption lithography[25,31,46]. The resulting structures are compared with the optimized geometry to ensure proper writing, as shown in Figs. 3a, b, displaying the experimentally fabricated structure and 3D computer model of the intended structure, respectively, both taken at the same tilt angle and estimated focal distance. The charging (white banding) in Fig. 3a is largely due to the undercut

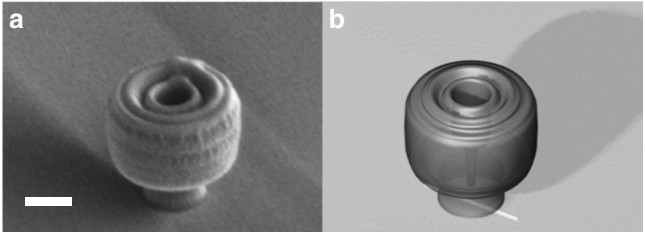

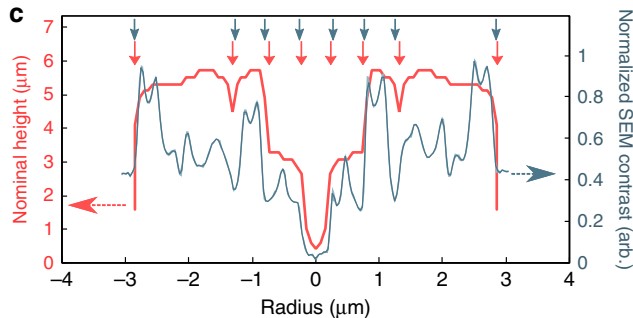

**Fig. 3** Fabricated nanophotonic lens. **a** SEM image of a nanophotonic lens structure printed over an embedded nanowire (not visible), and compared with, **b** computer 3D rendering of the as-designed structure, at the same imaging conditions. Scale bar = 2 μm. **c** Plot of the observed contrast in a top-view SEM image compared with the height profile of the as-designed structure, showing agreement between observed and designed features (indicated by the vertical arrows, at the onset of the step change in the data); the small shaded region is the standard deviation in SEM contrast, averaged over 10 pixels

present in the nanolens preventing uniform coating of the metal conduction layer applied before scanning electron microscopy (SEM) imaging. Additionally, the slight roughness on the nanolens is due to damage from the electron beam, developing during imaging. Finally, the detection contrast of a top-view calibrated SEM image is compared with the intended surface profile of the optimized geometry in Fig. 3c. This contrast arises from a change in topology of the nanolens, providing a useful metric for quantifying the lateral replication of the fabrication. Comparing eight of the steepest height changes in the design to the corresponding SEM contrast change (indicated by arrows) yields a root-mean-squared deviation between the two profiles of 6.9 nm, indicating the fabrication of the intended structure was indeed successful. It should be noted that the SEM contrast is not a direct profile of height, but simply should express contrast changes at areas of changing height. For additional details on all steps of the fabrication procedure, please see the Methods section.

**Measurement**. Measurements of the nanowire–nanolens systems are done using Fourier microscopy to image the angular distribution of broadband PL from the nanowires. The samples are positioned above an inverted microscope equipped with a 0.9 NA free-space objective, with both excitation and collection occurring through the nanolens structure[47]. As Fourier microscopy images the back focal plane of the objective lens, the intensity maps shown in Figs. 4a, b show the angular distribution of the broadband (750–900 nm) integrated emission, with the center corresponding to the surface normal and the dashed white line corresponding to the maximum collection angle (half-angle of 64.2°). The data are integrated using a Jacobian to transform the planar Cartesian pixel data to a hemispherical projection, yielding the total emitted power into the cone collected by the objective (defined by the 0.9 NA of the objective). The partial directivity of

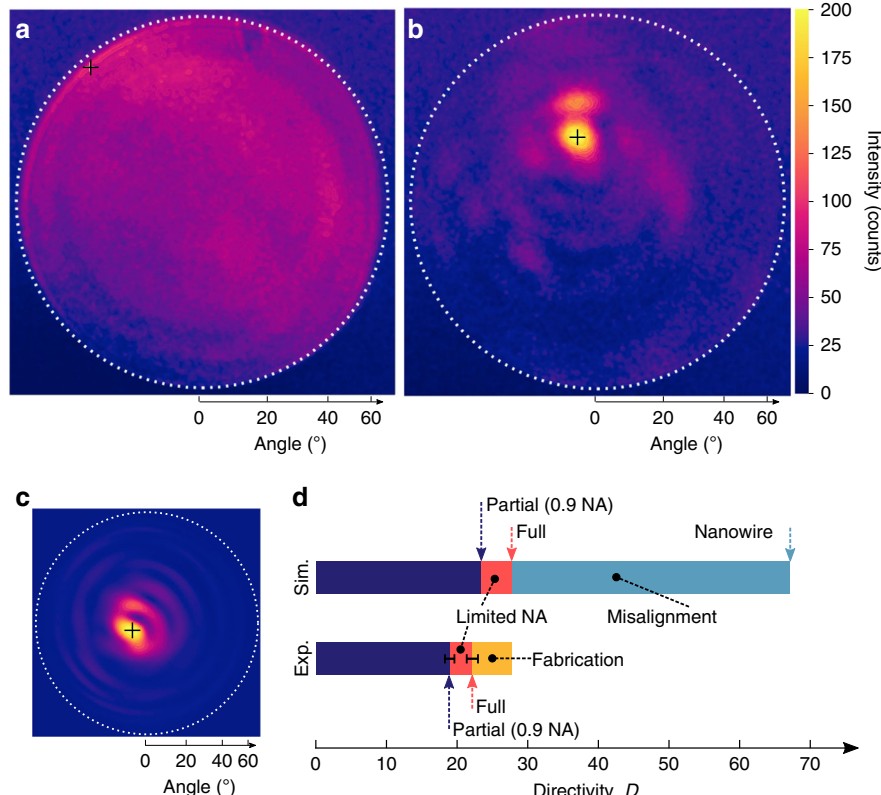

**Fig. 4** Fourier microscopy images of nanowire photoluminescence. **a** Angular distribution image of nanowire PL of a coated nanowire with no lens structure present, showing nearly homogeneous illumination, contrasted to **b** angular distribution image nanowire PL with a nanolens present, showing directional beaming into a half-angle of 3.5°. Dashed white lines represent the 0.9 NA objective collection area, and the black " + " indicates the maximum emission direction. **c** Simulated measurable (0.9 NA) angular distribution of the nanowire–nanolens system including the influences of misalignment and the finite source, beaming into a half-angle of 3.1°. **d** Attribution of likely mechanisms responsible for the difference between the experimentally measured and full simulated directivity, with arrows indicating the partial (within the measurable 0.9 NA) directivity and full directivity, along with the directivity of the nanolens with an ideal nanowire emitter

the measurement within this cone is then computed, using a 3 × 3 pixel average to determine the maximum emission direction (indicated by the small black "+" in Fig. 4) and radiated flux. The uncertainty of the measurement is largely dominated by noise, particularly at the low emission levels (e.g., when no lens is present), and so the variance of the surrounding region (outside the NA of the objective) is used to estimate the uncertainty in the measured directivity.

The enhanced directivity of the nanolens is demonstrated in Fig. 4. In Fig. 4a, we see the emission from a nanowire with no lens, appearing quite uniform throughout the angular space collectible by the objective, with slightly increased brightness at two extreme angles, and showing a low partial (observable) directivity of 2.2 ± 0.1, and no observed beaming, as is expected. The emission of the combined nanowire–nanolens system, however, shows a strong peak in emission near the center of the angular distribution image, representing a narrow beaming half-angle ($\sigma$) of 3.5 ± 0.4 degrees, and a partial directivity of 18.9 ± 0.7 within the observable NA. It should be noted that beaming into the absolute center of the plot is not entirely expected, as neither the alignment of the substrate to the optical axis of the measurement objective, nor the lens to the substrate surface (both from writing and development), were guaranteed to be perfectly orthogonal. Additionally, the position of the emitter under the nanolens can also induce some displacement of the emission peak, as discussed later. Together these likely account for the occurrence of the maximum emission slightly away from the substrate normal.

We can estimate the full directivity of the emission by first estimating the upper hemisphere (free space) partial directivity, extrapolating the emission for the remaining (unobserved) portion of the hemisphere from the furthest observable regions. Combined with the fact that the higher refractive index of the substrate as compared with free space (1.4 vs 1) causes a majority of the total emission to be propagated into the substrate (63%, from simulations), we thereby reach an experimentally estimated full directivity of 22.1 ± 0.5. Similarly, by performing this analysis for the nanowire without the presence of the nanolens, the full 3D directivity is computed to be 1.1 ± 0.1. Thus, through the application of the nanolens, the peak photon density is increased by a factor of 20.

## Discussion

Although the experimental results already show a great increase in directivity due to the nanolens, from 1.1 to 22.1 after application, it is instructive to understand the differences between the measurements and simulation in order to further improve future systems. Three significant factors differentiate our experimental measurements from the initial theoretical prediction: the finite emitter size, a potential misalignment of the lens to the emitter, and the limited NA of the measurement system.

We first include the contribution from the fact that the emitter in the experimental case is not a point dipole, but a finite nanowire. Although the emission comes largely from one (doped) end of the wire, the emission is still more extended than in the

theoretical (point dipole) case. This can be compensated for in the simulations by computing the far-field projection of point emitters at different displacements from the center of the nanolens, and summing together the intensity of an array of emitters representing the extended area of the nanowire. Using a Gaussian-weighted intensity distribution from the nanowire emitting region with a standard deviation ($\sigma$) of 240 nm (as measured from real space PL images of bare nanowires, with the system point spread function deconvoluted, see Methods), the expected theoretical directivity is reduced to the aforementioned value of 67.1.

It is likely that the alignment of the nanolens to the center of the emitting nanowire region is imperfect. A displacement of the nanolens orthogonal to the length of the nanowire of 300 nm, along with a displacement of the lens along the nanowire of 300 nm from the center of the emitting region yields a qualitatively similar angular emission profile to that of the experiment, as shown in Fig. 4c (see Methods for details). Including this displacement in the simulation gives a directivity of 27.6. This therefore represents the theoretically expected directivity of the physically produced system if the measurement and fabrication were perfect, in agreement with the experimental value of 22.1 ± 0.8. The beaming half-angle of the simulated system of 3.1° also agrees well with the experimental half-angle of 3.5°, as does the beam profile (see Methods). Finally, similar to the experimental calculation of the full directivity, we can additionally include in our simulations the NA-limited collection. Doing so, we compute that the expected partial directivity of the nanowire–lens structure to be 23.4 within an NA of 0.9, comparable to the experimental partial directivity of 18.9 ± 0.7.

The comparison between the experimental and simulated directivity shown in Fig. 4d gives an indication of both the accuracy of the simulation, as well as the fabrication quality of the nanolens structures. It can be seen that the imperfect alignment of the nanolens above the nanowire accounts for the largest reduction of the system directivity, followed closely by the use of a finite emission source. The remaining ~25% discrepancy between the simulated and experimental directivity is likely largely due to fabrication defects in the nanolens creation (e.g., the slight bulge at the top of the nanolens in Fig. 3a, from the closing of the rings that compose the nanolens structure), measurement error, and background noise in the experiment. The fact that the partial directivities agree somewhat better (~20% discrepancy), indicates that the experimental full directivity is likely actually even higher than measured, due to the propagation of the background intensity at the edge of the measurement into the full $4\pi$ steradian space. The relatively small discrepancy between the final simulated and experimental directivity values suggests both that fabrication was highly successful, and that the losses are well understood.

Herein we have shown that a simple EA can be used to design 3D nanophotonic lens structures showing theoretical directivity values of 101 for point emitters and 67 for nanowire emitters, with nearly constant response over a 200 nm wavelength range. This corresponds to a 3× increase in directivity, a 6× increase in F/B ratio, and a 1.7× increase in emission out of the substrate as compared with micro-spheres of the same radius. We have demonstrated the fabrication of such nanolenses, and combined the nanolenses with nanowire emitters. The directivity of the nanowire PL was shown to be increased by the nanolens from 1.1 to 22.1, and with a beaming half-angle of 3.5°, in good agreement with simulations. The simulations can also explain the reduction in the expected directivity, and attribute the largest reduction factors to the extended emission region of the nanowire, and imperfect alignment of the nanowire and nanolens. The full writing files are available as Supplementary Software 1 online,

allowing anyone with access to a Nanoscribe tool to quickly fabricate the same structures for themselves.

We expect that in future studies utilizing more confined nanostructure emitters (e.g., vertical nanowires, or quantum dot emitters) along with enhanced alignment techniques, even higher directivities of similarly produced nanolenses should be readily achievable. Additionally, by integrating back-reflectors or structured substrate surfaces, the attainable directivity, and out-coupling efficiency of such systems could be enhanced even further.

Finally, the methodology described herein is quite general—a combination of flexible 3D design with a virtually universally compatible fabrication technique, to create complex structures. Although laser lithography was used for our proof-of-concept demonstration, there has been great progress in making complex 3D dielectric shapes through scalable self-assembly processes[48], which even allow for conversion to light-emitting 3D structures[49]. These methods may thus also be possible to employ for the design and creation of structures for applications in spectroscopy, microscopy, photocatalysis, or any number of other processes where nanoscale control of light is demanded.

## Methods

**Evolutionary algorithm.** The EA in this work operates by dividing the space surrounding our fixed active element (e.g., nanowire, dipole emitter) into a 2D matrix of 40 × 40 pixels, with the optimization determining if each pixel should contain air ($n = 1$) or polymer material ($n = 1.45$). The breeding takes place by overlaying the two matrices to be bred, keeping regions where both matrices agree, and using a coarse random matrix (4 × 4 segments, with the size randomized to prevent preferential transition lines) to decide between regions where the values differ. The use of a coarse arbitration matrix prevents the creation of small single-pixel features in the regions of disagreement between the two structures, biasing the optimization toward producible structures. The matrix is finally smoothed through convolution with an asymmetric matrix estimating the point spread function of the two-photon lithography system (accounting for the lower vertical resolution of the process) before the simulation is run. After being simulated, results with performance exceeding the lowest of that in a pool of the best structures are included in the pool, replacing the previous lowest structure. For more details on the algorithm, see Supplementary Note 1.

**Simulations.** Simulations are performed using FDTD Solutions (Lumerical Inc.). During the optimization, reciprocity is leveraged in simulating the emission of the nanowire, in that the simulation is performed by calculating the absorption of a plane wave incident on the nanowire–lens system. Simulation details are provided in Supplementary Note 2.

After the design is complete, the full directivity is simulated, first by calculating the far-field emission of a dipole emitter located at the center of the nanowire. The transformation to the far-field is performed using a method based on reciprocity arguments in a modified version of a freely available software package[50]. This takes into account emission in all directions, and allows for the substrate interface (change in refractive index) to be taken into account when performing the transformation. Each of the three cardinal dipole orientations are calculated separately and their flux averaged when calculating the point emitter directivity. F/B ratios are calculated as the direct ratio of the emission maximum divided by the emission diametrically opposite.

Simulations are run over the full PL spectrum range (750–900 nm) for assessing the broadband response as shown in Fig. 2f. The single wavelength expressing a directivity equal to that of the PL-weighted response (840 nm) is used in the emission profiles in Figs. 2b, d. A plot of a 2D slices of the emission throughout the 750–900 nm range is shown in the Supplementary Figure 1. The time-domain plots in Fig. 2e are calculated for a short pulse (corresponding to a wavelength bandwidth of 194 nm) propagating through the system, at time delays of 25, 41, 58, and 66 fs from the beginning of the simulation. The robustness of the design to misalignment of the lens with respect to the substrate surface (into the substrate) is also analyzed, showing little change in the directivity for displacements up to 200 nm (see Supplementary Figure 2).

Finally, a series of simulations are run with dipole emitters displaced from the center of the nanowire–lens system, and a single far-field transformation in the upward direction is performed. The far-field flux is then combined incoherently, corresponding to positions of an extended source; in this case all points in a nanowire, with a Gaussian weighting chosen to replicate the emission from the nanowire PL observed in real space. This allows simulations of the experimental imaging conditions on sources with arbitrary sizes and orientations to be performed rapidly. The position of the simulated nanowire is simply adjusted to produce a profile similar to that observed in experiment. The fit is vetted through

the characterization of the beaming half-width, as well as comparing the profile of the major and minor peaks present in the two images (see Supplementary Figure 3).

The control simulation of the microsphere is performed similarly to those of the nanolens. The radius of the sphere is fixed to be the same as the nanolens, and the position is optimized to give the highest directivity, found to occur when the sphere is placed directly on top of the substrate (not submerged). A schematic and full emission profile is shown in Supplementary Figure 4.

**Fabrication**. The GaAs nanowires were grown in a metal-organic chemical vapor deposition system via the vapor–liquid–solid growth mechanism. The nanowire growth was catalyzed by 50 nm diameter Au nanoparticles. The nanowires were nucleated at 450 °C. The GaAs core and the aluminum gallium arsenide (AlGaAs) shell were grown at 375 °C and 750 °C, respectively. Trimethylgallium (TMGa), trimethylaluminium (TMAl) and arsine (AsH$_3$) were used as Ga, Al, and As sources, respectively. The nanowires have a measured diameter of ~80 nm, with a 50 nm GaAs core with a 15 nm thick AlGaAs passivation coating. The wires are drop-cast onto glass coverslips, and spin-coated with a ~200 nm thick SU-8 resist polymer layer, which is cured under ultraviolet illumination for 10 min. This layer prevents the need for the writing laser to expose on the nanowire directly, preventing damage to the wire.

To produce the nanolens structures, the optimized nanolens geometries are first translated into a series of concentric rings by an automated script, fitting the approximate empirical point spread function of the writing laser to the designed structure cross-section. The translated ring structures are then written directly in drop-cast OrmoComp (micro resist technology GmbH) resist (chosen for its low fluorescence, high transparency, and stability up to 270 °C)[51] via two-photon absorption lithography (Photonic Pro, Nanoscribe GmbH). Total writing time for one nanolens (including interface detection and adjustment) is 51 s. The structures are developed in mr-Dev 600 (micro resist technology GmbH) for 25 min, isopropyl alcohol for 5 min, and ethanol for 1 min. After development, the samples are dried using supercritical carbon dioxide. Care is taken to avoid agitation of the sample during development and drying, but occasionally nanolenes are observed to be tilted after the process is finished. During the development of the fabrication process, and after measurements, samples are sputter-coated with 5 nm of Cr and 10 nm of Au conduction layers, and imaged in a SEM (Verios 460, FEI Company). The lens profile and SEM contrast are compared by measuring the relative position to the center of the lens of the eight largest changes in height, along with the corresponding change in contrast of the SEM image, as measured from the step onset (upward transition).

The optimized structure investigated here utilizes the full width constraint of the optimization (5.7 μm), but only 5.8 μm of the 8.4 μm height limitation, and a solid volume of 89.6 μm$^3$, or 151.2 $\lambda^3$ for the PL peak (840 nm). The asymmetry in the optimization box is chosen to reflect the asymmetry in the lithographic point spread function.

**Measurement**. The measurement setup is similar to that described previously[47]. The sample is mounted over an inverted microscope objective (Nikon Plan Fluor 100 × 0.90 NA free-space objective), with excitation and collection occurring through the same single objective. The sample is excited with a focused 532 nm wavelength 500 ps pulsed laser (STG-03E-1S0, Teem Photonics), with a 2 kHz repetition rate, and with an approximate pulse energy of 2 nJ after attenuation by an ND 1.5 filter. The attenuation of the power is performed to reduce luminescence from the glass substrate or objective lens. A dichroic mirror helps to reduce propagation of the excitation beam into the collection pathway, and a 532 nm notch filter, and a 750 nm long-pass filter in the collection pathway further reduces non-PL emission from being imaged, while still collecting nearly the entire broadband PL spectrum. The PL is imaged using a cooled (−20 °C) CCD camera (Andor Clara) with a 20-s integration time, and repeated three times per nanowire measurement. Background measurements with the laser focused away from the nanowire are taken using the same laser power and CCD settings immediately after the nanowire measurement. Additional measurement details are provided in Supplementary Note 3.

Real space images of bare nanowire PL are used to determine the extent of the emitting region, and shown in Supplementary Figure 5.

**Analysis**. The data are processed by a custom script written in the Julia programming language[52]. This is done by first subtracting the background images from the nanowire measured data, and removing 12 dead (always off) pixels from the camera data. The observable region of the back focal plane is then fit using emission from bright point emitters, defining the maximum NA of the measurement. These data are thus a projection of the emission into a 64.2° half-angle cone, and is presented in the polar plots in Figs. 4a, b.

In order to calculate the directivity value, the data within the NA are stepped through, calculating the radial distance to each pixel, and creating a dataset of the mean radial response of the image. The radial response of both experimental measurements discussed here are shown in the Supplementary Figure 6. The radial response is then weighted by a projection to a spherical section, and the partial directivity is computed for the angular space within the NA. To compute the full directivity, the average emission in the 25% furthest region within the NA is

extrapolated to the remaining unobservable region of angular space, assuming the emission follows a Lambertian profile, giving the upper-hemispherical emission. This integrated emission is weighted by the ratio of emission into the substrate as calculated in simulations, used for the lower hemispherical emission, and the full directivity is thus computed.

**Code availability**. The Nanoscribe code for producing the nanolens structures investigated herein is available in Supplementary Software 1 and 2 online, containing the setup and coordinates for writing the structures, respectively.

## Data availability
The data that support the findings of this study are available from the corresponding author upon reasonable request

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

## Acknowledgements

We would like to thank E. Verhagen (AMOLF) for helpful discussions. We acknowledge Prof. S. Mokkapati (Cardiff University, Wales), and Profs. C. Jagadish and H.H. Tan for providing the nanowire samples through access of the Australian National Fabrication Facility, ACT node. This work is part of the research programme "Nanobricks: Building monocrystalline optoelectronics from welded nanocubes" with project number 14846, which is financed by the Netherlands Organisation for Scientific Research (NWO).

## Author contributions

E.J., S.A.M., and E.C.G. designed the study. E.J. performed the structure design, simulations and fabrication. E.J., S.K., and A.F.K. performed the experimental measurements. E.J. wrote the manuscript with input from all authors. E.C.G. and A.F.K. acquired funding for the work.

## Additional information

**Competing interests:** The authors declare no competing interests.

