## [Peer Review File · Nature Communications]

Reviewers' comments:

Reviewer #1 (Remarks to the Author):

From a technical point of view, this is really great work. The authors employed the approach based on the three-dimensional direct laser writing techniques to fabricate a collimator "lens". The optimized design looks quite complex but the performance is really superior. And apparently, a genetic algorithm optimization they employed here sounds like a perfect match for this fabrication technology. This paper does is a solid technological achievement, and a potentially big step forward for the laser writing technology.

Unfortunately, there is no much new physics here. But, from the other side, I recall similar papers published in Nature Photonics, such as A. Piggott et al, Inverse design and demonstration of a compact and broadband on-chip wavelength demultiplexer, Nature Photonics volume 9, pages 374–377 (2015) [Ref. 30 in the current manuscript].

Reviewer #2 (Remarks to the Author):

In this manuscript authors explain a methodology on how to design and fabricate lenses to engineer the directionality of a relatively isotropic point source into a directional one. The results have been illustrated with an example of a single GaAs nanowire but could be used in many other systems such as quantum emitters for quantum communication. The work is novel, and absolutely timely and important. For this reason i recommend the publication after the following comment has been taken into account:

One part of the experimental setting of the experiment is not clear. In fact, i deduce that the luminescence is excited by the same objective that collects it but I am not entirely certain. This point should be clarified. in addition, if this is the case, authors should also consider that excitation through this objective might change the intensity of collected light by increasing how much goes into the semiconductor. A real estimation on how much light is collected is needed. As an example, I would be interested to know if this objective also collects light that cannot be collected by commercial objectives (collection angle of max 70o). I would also like to know if this kind of methodology could be used to increase the spatial resolution of optical spectroscopy.

Reviewer #3 (Remarks to the Author):

This manuscript by Johlin et al. presents a 3D lens designed by evolutionary algorithm and fabricated by two-photon lithography for coupling light emission from a single nanowire to free-space with high directivity. Experimental characterization has been performed to validate theoretical predictions using Fourier microscopy. Controlling light emission and absorption from nanoscale semiconductors has been an active area of research in the past two decades or so, and here authors are taking a different direction by using a low-index dielectric lens to sculpt the emission from a single GaAs nanowire. Typically metallic and/or high dielectric materials are used to ensure compact, highly effective lenses and optical components. However, here reported device is quite bulky when compared with the size of the nanowire and therefore raising questions about the practicality of proposed approach. Although it makes sense to use such a directive lens for a single nanowire, authors also indicate potential applications such as photovoltaics which typically require significantly large area of semiconductor to reach desired efficiencies. Present approach by authors might fail to address such demands. However, I found the approach of using evolutionary design and two-photon lithography to build optical devices interesting. This manuscript can be

considered for publication in Nature Communications provided that authors address following comments.

- Authors should revisit their motivation and make sure that their proposed approach is feasible in terms of large-area fabrication for PV devices. Two-photon lithography is a rather slow process especially if fabricated structures are require ~ 100 nm resolutions or so.
- Authors mention 100 nm resolution limit for two-photon lithography. This is true for lateral dimensions however for vertical writing the resolution is around ~ 300 nm due to elliptical spot of focused laser beam. Did they take this into account in their design? They did not specifically mention this but end structure seem to have less variation in the out of plane (propagation) direction.
- Typically there will be fabrication issues resulting in not obtaining the ideal structure. They should include a cross-sectional SEM image to compare their design with fabricated structure. Normalized SEM contrast only provides information about the surface of the lens. Such non-idealities should also result in performance reduction. Also additional discussions on the robustness of their design to fabrication problems and temperature should be included.
- The term nanophotonic lens in the title is rather misleading. End device is rather bulky compared to conventional nanophotonic lenses such as plasmonic and high-dielectric antennas. Also proposed device is acting as an out-coupler rather than focusing incoming light to the nanowire therefore revisiting the term "lens" might also be considered.
- Fig. 2e presents field profile at different time frames, however instead of snapshots it will be more instructive for readers to see a movie. Therefore, I suggest that authors only plot the third or fourth field plot from top and refer to a movie in the supplementary info.
- Authors claim to have a broadband lens, however optical measurements are performed at a single wavelength which is the best operation wavelength. In order to support their claim, additional measurements at different wavelengths is required.

Reviewer #2 (Remarks to the Author):

In this manuscript authors explain a methodology on how to design and fabricate lenses to engineer the directionality of a relatively isotropic point source into a directional one. The results have been illustrated with an example of a single GaAs nanowire but could be used in many other systems such as quantum emitters for quantum communication. The work is novel, and absolutely timely and important. For this reason I recommend the publication after the following comment has been taken into account:

One part of the experimental setting of the experiment is not clear. In fact, I deduce that the luminescence is excited by the same objective that collects it but I am not entirely certain. This point should be clarified.

This is correct, the same objective excites the nanowire source and collects the emission from the nanowire, through the nanolens and into the objective. We agree that this is an important point to add, and have thus included additional information about the experimental measurement setup in the Methods section of the main text.

In addition, if this is the case, authors should also consider that excitation through this objective might change the intensity of collected light by increasing how much goes into the semiconductor. A real estimation on how much light is collected is needed. As an example, I would be interested to know if this objective also collects light that cannot be collected by commercial objectives (collection angle of max 70°).

We fully agree with the referee that the absolute intensity is likely enhanced both by the enhanced collection efficiency, and additionally by enhanced focusing of pump light into the semiconductor, thereby benefiting twice from the nanolens. It is well known that outcoupling efficiency and pump enhancement cannot be disentangled on basis of measurements of intensities alone - it is exactly for this reason that we base our claims of out-coupling efficiency on simulations. This allows us to rigorously keep track of exactly where all of the emission from the wire is going (including guided modes along the interface; emission into angles not collectable by the objective; and emission into the substrate). This point has been clarified in the Methods section where the weighting of the collected light is described.

The nanolens does indeed allow light to be collected that would not otherwise be observable by a commercial objective; this is partially indicated already through the 2x increase in light extraction from the substrate, which would be roughly equal to the improvement by the nanolens when used with an extremely high NA objective. We have additionally included a calculation of the increase in collection when using a more standard objective (NA of 0.7) once the nanolens is applied (a 2.8x increase). This new calculation is included as well in the main text of the manuscript.

I would also like to know if this kind of methodology could be used to increase the spatial resolution of optical spectroscopy.

This is a very interesting point; the nanolens can indeed focus illumination to a smaller point than a (e.g. 0.9 NA) free space objective, similar to a solid immersion lens. Simulations show a beam waist ($1/e^2$) diameter of $\sim 0.75 \lambda$ at the focal plane of the nanolens under planewave illumination, corresponding to an effective NA of ~ 1.14 , and thus could indeed allow for more selective excitation for higher resolution spectroscopy, at least when compared to that with a free-

space objective. This however was not the design criterion for the investigated nanolens, and so even better performance in this regard could be possible by following the same methodology but while designing for enhanced excitation localization. Accordingly, we have added an additional remark at the end of the conclusions of the main text, commenting on the additional possibilities for this methodology to be utilized for further applications.

Reviewer #3 (Remarks to the Author):

This manuscript by Johlin et al. presents a 3D lens designed by evolutionary algorithm and fabricated by two-photon lithography for coupling light emission from a single nanowire to free-space with high directivity. Experimental characterization has been performed to validate theoretical predictions using Fourier microscopy. Controlling light emission and absorption from nanoscale semiconductors has been an active area of research in the past two decades or so, and here authors are taking a different direction by using a low-index dielectric lens to sculpt the emission from a single GaAs nanowire. Typically metallic and/or high dielectric materials are used to ensure compact, highly effective lenses and optical components. However, here reported device is quite bulky when compared with the size of the nanowire and therefore raising questions about the practicality of proposed approach. Although it makes sense to use such a directive lens for a single nanowire, authors also indicate potential applications such as photovoltaics which typically require significantly large area of semiconductor to reach desired efficiencies.

Present approach by authors might fail to address such demands.

However, I found the approach of using evolutionary design and two-photon lithography to build optical devices interesting. This manuscript can be considered for publication in Nature Communications provided that authors address following comments.

- Authors should revisit their motivation and make sure that their proposed approach is feasible in terms of large-area fabrication for PV devices. Two-photon lithography is a rather slow process especially if fabricated structures are require ~100 nm resolutions or so.

We should note that photovoltaic improvement is simply one of several motivations for this work, and not its primary focus. However, we are very interested in pursuing applications for PV and are happy to provide the Referee with a further explanation of our vision of this application route:

The Referee quotes large-area fabrication throughput as a constraint. In our view, the challenges of sequential writing in resist can be overcome by established larger area techniques for 3D patterning of polymers; e.g. multi-layer soft imprint lithography (see Nanotechnology 24, 045304), or through laser holography (see Nature 404, 53). Alternatively, larger structures could be designed using the same methodology explored here, to be fabricated more quickly through single-photon absorption stereolithography, which can be more readily parallelized. Finally, we are in preliminary stages of a collaboration developing these structures via a completely bottom-up self-assembly technique which has demonstrated excellent control over 3D shape with transparent metal carbonates, and that can even be converted to light emitting structures. A mention of the final option here has been added to the Conclusions in the main text of the manuscript.

We furthermore note that the Referee mainly points to grid-scale PV applications. In our view, grid-scale PV is likely not the first application area of the proposed techniques. Rather, we envision micro- and nanostructured PV to present a more realistic short-term application. Here the aim is not necessarily to maximize efficiency, but rather to maximize the optical cross section of a device with a small active region. In this work, the nanolens increases the absorption cross-section of the nanowire by a factor of approximately 32. While this is admittedly smaller than the ratio of the projected area of the nanolens to that of the nanowire, it does make the “bulk” of the nanolens that the Referee refers to far less of an issue. Furthermore, if one accounts for the fact that nanowires often have an active region which accounts for most of the power generation, as well as the need for contacts taking up some area in horizontal devices, the nanolens focusing more photon density on the active area could indeed result in significantly higher active absorption cross section from a system including the nanolens structures relative to a horizontal nanowire device forgoing them. The motivation of “nanostructured photovoltaics” has been clarified in the introduction of the main text, and the increased absorption cross section information has been added to the Simulation section of the Supplementary Information file.

- Authors mention 100 nm resolution limit for two-photon lithography. This is true for lateral dimensions however for vertical writing the resolution is around ~300 nm due to elliptical spot of focused laser beam. Did they take this into account in their design? They did not specifically mention this but end structure seem to have less variation in the out of plane (propagation) direction.

This is an excellent point, and one we did indeed address in our design, but agree is insufficiently clear in the manuscript text; we have now clarified the point that the PSF during the optimization was asymmetric to account for the lower resolution in orientation of the optical axis, both in the main text as well as the Methods section.

- Typically there will be fabrication issues resulting in not obtaining the ideal structure. They should include a cross-sectional SEM image to compare their design with fabricated structure. Normalized SEM contrast only provides information about the surface of the lens. Such non-idealities should also result in performance reduction. Also additional discussions on the robustness of their design to fabrication problems and temperature should be included.

We agree that fabrication robustness is highly important. Indeed, Figures 4C and 4D are devoted to estimating the fabrication quality and its impact on optical performance. As displayed by the gold bars in 4D, simulations predict that the fabrication error accounts for the directivity reduction from 27.6 (in simulation of perfectly fabricated structures) to 22.1 (as calculated for the experimental measurements).

It is true that the comparison with the SEM contrast does only show the relative positions of the lens features in-plane, and it is for this reason that we include the comparison between the full 3D rendered as-designed nanolens, compared to the 50 degree tilt SEM image. We should note that the rendering is carefully produced with the same angle and focal distance as the SEM image, and so while not entirely quantitative, direct comparisons of features between the two images can still be made. This point had not been mentioned previously in the text, and so we have included this information now.

Furthermore, we have included an additional plot in the supporting information, showing the calculated response of directivity to vertical misalignment of the lens structure to a point emitter into the substrate, showing little change in directivity for misalignments up to 200 nm. This information along with a reference to the new SI figure have been added to the Methods section of the main text.

Thermal expansion or contraction of the lens should be equivalent to a slight change in the wavelength of operation, and due to the nearly uniform performance over a wide bandwidth, we would expect the lens to be quite thermally stable. The temperature stability should thus simply be determined by the stability of the OrmoComp material of which the nanolens is composed. The manufacturer cites a stability of up to 270 degrees C, and so we believe the nanolenses should be stable under most reasonable operating conditions. This stability information has also been added to the Methods section of the main text.

- The term nanophotonic lens in the title is rather misleading. End device is rather bulky compared to conventional nanophotonic lenses such as plasmonic and high-dielectric antennas. Also proposed device is acting as an out-coupler rather than focusing incoming light to the nanowire therefore revisiting the term “lens” might also be considered.

We believe the existence of critical features within the lens being significantly sub-wavelength (particularly for optical wavelengths) makes the term “nanophotonic” appropriate. This use of the word “nanophotonic” is in line with the field of nanophotonics as a whole; for instance microcavities, metasurfaces, and photonic crystals all derive their function from nanoscale critical dimensions, but overall sizes exceeding microns.

Additionally, while the lens does improve out-coupling from the substrate, the main influence is in changing the angular distribution of light to/from the active element. It can thus be seen as either focusing light from the far field onto the active element (nanowire or dipole) or performing the reciprocal process (converting emission from the localized active element into a single direction). This is indeed the function of lenses, and so we view this terminology also appropriate.

To address the referees concern, we have clarified the introduction where we focus on the difference between nanophotonic and micro-lens structures.

- Fig. 2e presents field profile at different time frames, however instead of snapshots it will be more instructive for readers to see a movie. Therefore, I suggest that authors only plot the third or fourth field plot from top and refer to a movie in the supplementary info.

This is an excellent idea – a corresponding video file has been included as a supporting online file, and we have included a reference to said video in the text.

- Authors claim to have a broadband lens, however optical measurements are performed at a single wavelength which is the best operation wavelength. In order to support their claim, additional measurements at different wavelengths is required.

We apologize for the confusion; the experimental measurements were not performed at a single wavelength, but are the fully broadband response over the entire PL spectrum of the nanowires. All experimental results are thus already broadband, representing wavelengths ranging from 750

to 900 nm. Single wavelength measurements would likely appear indistinguishable, as our simulations predict little change in lensing performance across this bandwidth. This has been clarified at two points in the main text of the manuscript as well as in the Methods section.

REVIEWERS' COMMENTS:

Reviewer #2 (Remarks to the Author):

This is a fantastic and extremely innovative paper. The authors have addressed all the questions, I suggest that it is published as is.

Reviewer #3 (Remarks to the Author):

I have read authors' response letter to reviewers and revised manuscript carefully. They have addressed all of my concerns and significantly increased the quality and strength of their manuscript. I strongly suggest publication of this manuscript in Nature Communications without further revisions.